# HMNet: Hierarchical Multi-Scale Brain Tumor Segmentation Network

**DOI:** 10.3390/jcm12020538

**Published:** 2023-01-09

**Authors:** Ruifeng Zhang, Shasha Jia, Mohammed Jajere Adamu, Weizhi Nie, Qiang Li, Ting Wu

**Affiliations:** 1School of Microelectronicss, Tianjin University, Tianjin 300072, China; 2School of Electrical and Information Engineering, Tianjin University, Tianjin 300072, China; 3Department of Cardiopulmonary Bypass, Chest Hospital, Tianjin University, Tianjin 300072, China

**Keywords:** brain tumor segmentation, multi-scale, depthwise separable convolution, conditional channel weight

## Abstract

An accurate and efficient automatic brain tumor segmentation algorithm is important for clinical practice. In recent years, there has been much interest in automatic segmentation algorithms that use convolutional neural networks. In this paper, we propose a novel hierarchical multi-scale segmentation network (HMNet), which contains a high-resolution branch and parallel multi-resolution branches. The high-resolution branch can keep track of the brain tumor’s spatial details, and the multi-resolution feature exchange and fusion allow the network’s receptive fields to adapt to brain tumors of different shapes and sizes. In particular, to overcome the large computational overhead caused by expensive 3D convolution, we propose a lightweight conditional channel weighting block to reduce GPU memory and improve the efficiency of HMNet. We also propose a lightweight multi-resolution feature fusion (LMRF) module to further reduce model complexity and reduce the redundancy of the feature maps. We run tests on the BraTS 2020 dataset to determine how well the proposed network would work. The dice similarity coefficients of HMNet for ET, WT, and TC are 0.781, 0.901, and 0.823, respectively. Many comparative experiments on the BraTS 2020 dataset and other two datasets show that our proposed HMNet has achieved satisfactory performance compared with the SOTA approaches.

## 1. Introduction

Glioma is the most common primary brain tumor, accounting for 70% of malignant primary brain tumors in adults. The World Health Organization (WHO) classifies brain tumors into grades I–IV [1], with different survival times for different grades; the higher the grade of a brain tumor, the more aggressive it is, and the shorter the survival time of patients [2], so the early diagnosis and treatment of brain tumors are crucial. Physicians usually use magnetic resonance imaging (MRI) [3] as the main basis for the clinical diagnosis of brain tumors. The four common MRI modalities of the brain include: T1-weighted (T1), contrast-enhanced T1-weighted (T1c), T2-weighted (T2), and fluid-attenuated inversion recovery (FLAIR) [4]. However, brain tumors are characterized by different morphologies and variable locations [5], and manual segmentation requires doctors to rely on their professional knowledge and work experience to analyze the condition, so human factors such as fatigue, memory, and lack of work experience may make the diagnostic results wrong during the segmentation process. Medical imaging and computer-aided diagnosis (MICAD) technology has been made possible by the growing connection between computer science and medicine. Combining MICAD technology with computer vision to allow doctors to see the results of MRI image segmentation and use them as a reference can speed up diagnostics and make them much more accurate.

Traditional brain tumor segmentation algorithms include threshold-based, clustering-based, atlas-based, and supervision-based algorithms [6,7,8,9,10]. Although the computational complexity of brain tumor image segmentation algorithms based on traditional machine learning is low, manually designed features cannot effectively utilize the rich information of MRI brain images. Moreover, such algorithms usually need to compute many features to ensure accuracy, which slows down the computation speed and increases the memory overhead.

In recent years, convolutional neural networks (CNNs) have been successfully applied in the field of medical images [11]. Unlike traditional machine learning methods, CNNs use massive amounts of data to learn representative features automatically. This way, they do not have to go through the complex process of extracting features. CNNs have been applied successfully in medical image classification [12], detection [13], segmentation [14,15], and other image processing tasks [16]. Many studies on brain tumor segmentation have used CNN-based methods, which have greatly improved the speed and accuracy of diagnosing brain tumors. Havaei et al. [17] designed a cascaded dual-path CNN structure. Convolution kernels of different sizes were used to learn local and global features of images simultaneously; then local features and global features were fused to obtain more context information. Salehi et al. [18] proposed an auto-context-based CNN algorithm to obtain the local features and global features using an approach based on three parallel two-dimensional convolutional paths in the axial, coronal, and sagittal directions. However, these algorithms are subject to necessary post-processing as they are executed based on image blocks. Ronneberger et al. [19] proposed the encoding-decoding network U-net, which increases the number of upsampling layers compared with the traditional FCN [20] and gradually restores the image details and image resolution, and the feature map concatenation of the encoding path and the decoding path can maintain complete contextual features. U-net has become the most popular basic network in medical image processing tasks. Researchers have improved U-net and produced variants of higher accuracy [21,22]. Marcinkiewicz et al. [23] proposed an improved structure of 2D U-Net, where MR images of three modalities are input into three independent encoders. Then, the feature channels are fused and fed into the encoders. Although 2D CNN-based brain tumor segmentation methods are efficient, 3D MRI image continuity and complete 3D spatial contextual information are difficult to capture. The 3D U-net [24] network extended the U-net architecture by replacing all 2D operations with their 3D operations. Mehta et al. [25] proposed a 3D U-net brain tumor segmentation network using 3D convolution. Myronenko et al. [26,27] reconstructed the input image by adding variational auto-coding branches to regularize the shared encoder and achieved first place in the BraTS 2018 and BraTS 2019 competitions, respectively. Attention mechanisms are often introduced in brain tumor segmentation networks to make the model more focused on tumor-related regions and improve segmentation performance [28,29,30,31]. Cao et al. [15] used a novel multi-branch 3D shuffle attention module as the attention layer in the encoder, which grouped along the channel dimension and divided the feature maps into small feature maps.

Although the segmentation networks based on encoding and decoding structures can achieve satisfactory performance, continuous upsampling does not bring back image detail information from multiple downsampling, cannot bring back spatial information well, and does not pay much attention to edges. The extracted feature approach inevitably produces a fuzzy feature mapping after multiple convolutions and loses some important details, such as the boundaries of the brain tumor. Sun et al. [32] proposed a high-resolution network (HRNet). HRNet has been successfully applied in many tasks, such as human pose estimation [33] and semantic segmentation [34]. Compared with U-Net, it can maintain accurate spatial feature information due to the high-resolution branch. However, a large number of repetitive inter-fusion operations between multiple stages increase computational complexity, and intensive feature fusion computes a large amount of redundant feature information.

Although 3D convolution has good segmentation performance, it significantly increases the number of model training parameters, which results in a larger computational overhead. Nowadays, in edge computing systems, model efficiency is also an important aspect to be considered. With the rapid development of CNN technology, many lightweight networks [35,36,37] have been proposed to further reduce the model complexity and the memory occupied. Ma et al. [38] utilized three operations, channel split, depthwise separable convolutions, and channel shuffle, to enable the model to greatly reduce the computational effort of model while maintaining accuracy. Chen et al. [39] exploited the advantage of separable 3D convolutions to reduce the network complexity but with much lower segmentation accuracy. To balance network efficiency and accuracy in 3D MRI brain tumor segmentation, Chen et al. [40] proposed a new 3D expanded multi-fiber network (DMFNet) that uses efficient group convolution on top of multi-fiber units and obtains a multi-scale image representation for segmentation by introducing weighted 3D expanded convolution operations, which achieved precise segmentation while reducing the number of network parameters. However, the problem of the lack of exchange of channel information was not solved. Zhou et al. [6] proposed an efficient 3D residual neural network (ERV-Net) by first utilizing the computationally efficient network 3D ShuffleNetV2 as an encoder to reduce GPU memory and improve the segmentation efficiency of the network, and then introducing a decoder with residual blocks to avoid degradation.

The challenge of this study is to design a 3D brain tumor segmentation network with high accuracy and efficiency. The networks based on the U-net network ignore the gap problem between low-level visual features and high-level semantic features, have limited ability to reconstruct spatial information of brain tumors, and pay little attention to boundaries. The purpose of this paper is to achieve fast and accurate segmentation of brain tumors using a high-resolution-based network.

In this paper, we propose a hierarchical multi-scale brain tumor segmentation network (HMNet) to segment brain tumors with high efficiency and accuracy. Firstly, the network contains a high-resolution branch to maintain accurate spatial feature representation and multi-resolution branches to acquire multi-scale receptive fields and multi-scale features, which helps to segment brain tumors of different sizes and shapes. Secondly, to reduce the network parameters and computation to improve the efficiency of brain tumor segmentation, we exploit a 3D shuffle block [38] for feature extraction, then propose a lightweight conditional channel weighting (LCC) block to improve it. Thirdly, we propose a lightweight multi-resolution feature fusion (LMRF) module to further reduce model complexity and reduce feature redundancy. Finally, we assess our experimental results using the BraTS 2018, BraTS 2019 and BraTS 2020 datasets.

The contributions of this paper can be summarized as follows.

We propose a hierarchical multi-scale brain tumor segmentation network (HMNet). HMNet uses a parallel multi-resolution feature extraction network with a high-resolution branch and parallel multi-resolution branches to figure out what brain tumors look like.We propose a lightweight conditional channel weighting (LCC) block based on a 3D shuffle block [38] to overcome the computational problem caused by traditional 3D convolution and enhance useful features.We propose a lightweight multi-resolution fusion (LMRF) module to solve the problem that the traditional method of downsampling and upsampling has a lot of computational complexity.

The remaining sections of this article are structured as follows. Section 2 describes our methodology in depth. Section 3 contains the appropriate experimental data and analyses. Section 4 discusses the effectiveness of our proposed network. Section 5 summarizes the work of this paper.

## 2. Method

According to the above motivation, a hierarchical multi-scale brain tumor segmentation network (HMNet) is proposed. The architecture of HMNet is shown in Figure 1 and Table 1. HMNet includes three important components: (1) a backbone based on the parallel multi-resolution feature extraction network (PMRNet). It can maintain high-resolution feature extraction, and multi-resolution features are exchanged and fused to adapt the receptive field to brain tumors of different sizes and shapes; (2) the lightweight conditional channel weighting (LCC) block. It can overcome the large computational overhead by introducing depth-wise separable convolution. Then, we take advantage of multi-resolution and spatial information with a cross-resolution weighting unit (CWU) and spatial weighting unit (SWU) to improve network performance; and (3) the lightweight multi-resolution fusion (LMRF) module. It can overcome the large computational overhead caused by the fusion layers and reduce the redundancy of the feature maps.

### 2.1. Parallel Multi-Resolution Feature Extraction Network (PMRNet)

Inspired by the high-resolution network (HRNet) [32], a parallel multi-resolution feature extraction network (PMRNet) is proposed, as shown in Figure 1.

The PMRNet consists of four stages (Stage 1–Stage 4). Each new stage (except Stage 4) adds a new branch to the previous one, which has half the resolution and twice the number of channels of the previously added branch. Every stage consists of a lightweight conditional channel weighting (LCC) block to fully exploit multi-resolution features and a lightweight multi-resolution fusion (LMRF) module to aggregate rich multi-resolution contextual information. The feature maps with different resolutions are generated in parallel by PMRNet, which can maintain high-resolution feature extraction, and multi-resolution features are exchanged and fused to adapt the receptive field to brain tumors of different sizes and shapes. In this paper, we designed four subnetworks with different resolutions, 64 × 64 × 64, 32 × 32 × 32, 16 × 16 × 16, and 8 × 8 × 8 from top to bottom.

The PMRNet has three advantages. Firstly, the PMRNet can preserve the details of brain tumors due to the high-resolution branch. Next, the multi-resolution branches can provide different receptive fields for brain tumors with different sizes. Moreover, the exchange and fusion of the multi-resolution features can extract more abundant tumor features.

### 2.2. Lightweight Conditional Channel Weighting (LCC) Block

The 3D brain tumor segmentation task is usually time-consuming and GPU memory-consuming, so we propose an efficient unit, a lightweight conditional channel weighting (LCC) block, as shown in Figure 2 and Table 2. We extended the shuffle block [38] to the 3D version and improved it to extract features in PMRNet. First, the input channels were split into two branches by the operation called “Channel Split”. Then, the main branch ran a channel weighting unit (CWU), 3 × 3 × 3 depthwise separable convolution (DWConv), and spatial weighting unit (SWU). The CWU and SWU replace the costly traditional 1 × 1 × 1 convolution in the 3D shuffle block [38] to overcome the problem that the heavily-used 1 × 1 × 1 convolution becomes computational. The other branch was mapped as constant. The output feature maps of the two branches were merged by channel-wise concatenation, and a channel shuffle was performed on the merged feature maps. This operation takes a channel-wise concatenation approach. The input feature map was first divided into several subgroups by channel and handed over to different convolution kernels for group convolutions. Then, different subgroups were randomly extracted for channel shuffle into a new feature map so that the input feature information from different groups was fused by the next group convolution. This can enhance the exchange of information between channel groups and ensure that the input and output channels are fully correlated with each other.

As shown in Figure 3, we used a cross-resolution weighting unit (CWU) instead of the first 1 × 1 × 1 convolution block in the shuffle block [38]. Firstly, the input feature map of the parallel multi-resolution branch was subjected to the adaptive average pooling (AAP) operation to make its resolution equal to the minimum resolution of the branch and concatenated in the channel direction. Global average pooling (GAP) was performed to compress the spatial features of the input feature map, and the compressed feature maps were subjected to a one-dimensional convolution operation with k kernel size to learn the importance of different channels. The value of *k* varied with the number of channels (*C*). The relationship between them can be written as [41]:(1)$$k=|\frac{\log_{2}\left(C\right)}{2}+\frac{1}{2}|.$$

Then, the weight maps obtained by the sigmoid operation were upsampled in order to make the resolution of the weight maps consistent with the resolution of the feature maps of their corresponding branches, so the weight of a particular position in a branch fused with the multi-resolution feature.

We also introduced a spatial weighting unit (SWU). For each resolution in PMRNet, we computed the spatial weights, which are homogeneous to spatial positions. The process is implemented as:(2)$$\begin{array}{c} X\to \mathrm{GAP}\to \mathrm{Conv}1\times 1\times 1\to \mathrm{ReLU}\\ \to \mathrm{Conv}1\times 1\times 1\to \mathrm{sigmoid}\to W, \end{array}$$
where X denotes the input feature map, and W denotes the output weight map. The global average pooling (GAP) operator gathers the spatial information of brain tumors from all positions. By weighting the channels with the spatial weights, each element in the output channels receives the contribution from all the positions of all the input channels.

### 2.3. Lightweight Multi-Resolution Fusion (LMRF)

The fusion layer between stages in PMRNet is used for information interaction between multi-resolution branches. The traditional method is to use 3 × 3 × 3 convolutions with stride = 2 to downsample to reduce the resolution and use the nearest neighbor upsampling to increase the resolution, then use 1 × 1 × 1 convolution to implement dimensionality reduction. In order to avoid the high computational complexity problem caused by the fusion layer, we propose a lightweight multi-resolution fusion (LMRF) module. The structure of the LRMF module is shown in Table 3. We use a depthwise separable convolution (DWC) of 3 × 3 × 3 with stride = 2 to downsample. To solve the computationally intensive problem of a 1 × 1 × 1 convolution when increasing the resolution, we proposed a method that is similar to the Ghost module [42] to implement the same function as a 1 × 1 × 1 convolution. The structure is shown in Figure 4, firstly using a 1 × 1 × 1 convolution to generate a feature map with the number of channels as half of the number of output channels, and then the feature map of the required number of channels is generated by DWC and channel-wise concatenation. This method can achieve the same function as a 1 × 1 × 1 convolution and reduce feature redundancy.

## 3. Results

### 3.1. Dataset and Pre-Processing

We evaluate our HMNet on three datasets [43,44,45]: (1) the BraTS 2018 dataset. It includes a training set of 285 samples and a validation set of 66 samples. (2) The BraTS 2019 dataset. It includes a training set of 335 samples and a validation set of 125 samples. (3) The BraTS 2020 dataset. It includes a training set of 369 samples and a validation set of 125 samples. Every sample in the training set includes four MRI modality images and a ground truth label, as shown in Figure 5. The size of the MRI images is 240 × 240 × 155 mm^3^, with the spacings among voxels are 1 × 1 × 1 mm^3^. The ground truth label includes enhancing tumors (label 4), peritumor edema (label 2), and necrotic and non-enhancing tumor cores (label 1). The validation results need to be uploaded to the online platform (https://ipp.cbica.upenn.edu/, accessed on 1 December 2022) for validation. The segmentation results can be divided into three classes: ET (label 1), WT (labels 1, 2 and 4), and TC (labels 1 and 4).

To facilitate model training, we preprocessed the brain MRI images to address the noise and grayscale inhomogeneity. First, we standardized MRI brain regions to reduce the influence of intensity inhomogeneity. The strategy of data augmentation was used to expand the training set in order to prevent the overfitting problem: (1) random mirror flips in the axial, coronal, and sagittal directions with a probability of 0.5; (2) random rotations in the range of (−10°, 10°).

### 3.2. Evaluation Metrics

The performance of HMNet can be evaluated quantitatively using computational complexity and segmentation accuracy. The Dice similarity coefficient (Dice) and the 95% quantile of Hausdorff distance (Hausdorff95) are used to evaluate the segmentation accuracy. Dice is a measurement of the similarity between the brain tumor segmentation result and the ground truth. Hausdorff distance (HD) is used to measure the distance between the brain tumor segmentation regions and the ground truth. Dice and HD are defined as [6]:(3)$$\mathrm{Dice}=\frac{2TP}{FP+2TP+FN}$$
(4)$$\mathrm{HD}=d\left(P,T\right)=\max\left\{\sup_{p\in P}\inf_{t\in T}d\left(p,t\right),\sup_{t\in T}\inf_{p\in P}d\left(p,t\right)\right\}$$
where FP, TP, and FN denote the number of false positive, true positive, and false negative voxels, respectively. *P* denotes the pixel set of the predicted tumor region, and *p* denotes the pixel in the set. *T* denotes the pixel set of the ground truth, and *t* denotes the pixel in the set. d(p,t) is the distance between pixel points. sup denotes the supremum of tumor region, and inf denotes the infimum of tumor region. The larger the Dice is, the better the segmentation result of the network is. The smaller the Hausdorff95 is, the closer the distance between the predicted brain tumor boundary and the ground truth boundary is, and the higher the segmentation quality is. Parameters (Params) and floating point of operations (Flops) were used to evaluate the computational complexity of the network. Params represents the spatial complexity of the network, and Flops represents the time complexity of the network. They are expressed as:(5)$$\mathrm{Params}=k_{d}\times k_{h}\times k_{w}\times C_{in}\times C_{out}$$
(6)$$\mathrm{Flops}=2\times \left(k_{d}\times k_{h}\times k_{w}\times C_{in}\right)\times C_{out}\times d\times h\times w$$
where $k_{d}$, $k_{h}$ and $k_{w}$ denote the depth, height and width of the convolution kernel, respectively. $C_{in}$ and $C_{out}$ denote the number of input and output channels, respectively. *d*, *h* and *w* denote the depth, height and width of the image, respectively.

### 3.3. Implementation Details

In this paper, the experimental code was carried out in Python 3.6, the server environment was Ubuntu 16.04, the CPU was Intel Core i9-9900X (3.5HGz), the graphics card was an Nvidia GTX2080Ti with 11 GB, and all the experimental networks were built using a PyTorch framework. The experimental parameters were set as follows: the network input image size was 128 × 128 × 128, the batch size was 2, and the maximum number of training epochs was 500. The Adam optimizer was used for the automatic optimization of the network, and the initial learning rate was $10^{-4}$. During training, we divided the training set of every dataset into five parts, four of which were used as the local training set, and the other one was used as the local validation set. The code of this paper is available at https://github.com/jia-604/HMNet, accessed on 1 December 2022.

### 3.4. Comparison with Non-Lightweight Brain Tumor Segmentation Networks

To verify the performance of the proposed network, we conducted comparison experiments between the proposed network and current non-lightweight brain tumor segmentation networks on the BraTS 2020 dataset. The selected non-lightweight comparison networks are the 3D U-net [24], the NoNew-Net [46], the cascaded 3D densely connected U-network-based brain tumor segmentation network proposed by Ghaffari et al. [47], the Transformer-based brain tumor segmentation network proposed by Wang et al. [48], and the dResU-Net proposed by Raza et al. [49].

The results in Table 4 show that the proposed HMNet has the advantage of low model complexity compared to the non-lightweight network. Compared with the traditional 3D U-net, the number of parameters of our network was reduced to a fifth, and the amount of computation was reduced by 1540.1 G. The Dice of HMNet were 7.5%, 3.9%, and 8.6% higher than 3D U-net in ET, WT, and CT, respectively, which means that our network is more accurate than 3D U-net for brain tumor segmentation. Figure 6 shows more obviously that our network greatly improved segmentation accuracy compared to 3D U-net. It can be observed in Figure 7 that our network can extract more detailed information than 3D U-net. Table 4 shows that our network can achieve excellent segmentation results in WT and TC compared with the other non-lightweight network. Therefore, the HMNet is competitive in non-lightweight networks.

### 3.5. Comparison with Lightweight Brain Tumor Segmentation Networks

To verify the efficiency of our proposed network, we compared the performance of HMNet and other high-performing lightweight networks. Table 5 and Figure 7 display the experimental results. We retrained the network in Table 5 on the BraTS 2020 dataset.

Table 5 shows that the HMNet uses only about a quarter of the parameters of 3D ESP-Net, while the Dice of ET, WT, and TC are improved greatly, increasing by 9.1%, 3.0%, and 3.7%, respectively. Compared with DMF-Net, HMNet has 79.4% fewer parameters, while the Dice coefficients of WT and TC have improved by 0.2% and 0.6%, respectively. HDC-Net is currently the lightest brain tumor segmentation network, but the network is so light that it leads to poor segmentation metrics. Compared with HDC-Net, the Dice coefficients of ET, WT, and TC of the segmentation results of HMNet improved by 1.4%, 0.50%, and 2.60%, respectively, and the Hausdorff distances of ET and TC were shortened by 11.1 mm and 7.1 mm, respectively. As we can see in Figure 7, our network is more accurate than other networks in segmenting brain tumor boundaries. The above analysis shows that the proposed network is competitive in lightweight networks, and the proposed improvement direction is feasible.

### 3.6. Analysis about the Baseline

We designed a baseline network in which the PMRNet uses an ordinary bottleneck block in HRNet [32] for feature extraction and used ordinary upsampling and downsampling for multi-resolution fusion. In order to evaluate the effect of the number of Bottleneck blocks on the network segmentation performance, we set the number of bottleneck blocks as 1, 2, 3, and 4 in the same experimental environment, corresponding to network 1, network 2, network 3, and network 4, respectively. It can be seen from Table 6 and Figure 8 that as the number of blocks increased, the number of parameters and computation increased, but the improvement in segmentation accuracy was not significant, and even the accuracy of TC decreased. Considering both the network complexity and segmentation accuracy, we finally set the number of feature extraction blocks for each branch of PMRNet to 1.

In order to evaluate the impact of the input size of PMRNet and the Skip operation on the segmentation results, we designed a series of experiments, the results of which are shown in Table 7. The experimental results show that the PMRNet input size of 64 × 64 × 64 obtained better segmentation results, and the Skip operation improved the segmentation Dice of ET regions significantly. There was no decreasing trend in the average Dice of the three tumor regions, which proves the effectiveness of the Skip operation.

### 3.7. Ablation Experiments

We conducted ablation experiments to evaluate the effectiveness of each module in HMNet on the BraTS 2020 dataset. The results of the ablation experiments are shown in Table 8 and Figure 9. We first used the shuffle block to replace the ordinary bottleneck block as Network 2 in Table 8, which reduced the network parameters by 86% and the computation by 30.2 G. The Dice coefficient of the ET was reduced by 1.1%, but the Dice coefficients of the WT and CT were 0.3% and 0.6% higher, respectively. Therefore, the shuffle block can guarantee no decrease in accuracy based on greatly reducing the network complexity. To evaluate the effectiveness of the LCC block, we used the LCC block to replace the shuffle block as Network 4 in Table 8, which further reduced the parameters and computation while improving the segmentation accuracy of ET, WT, and CT by 0.9%, 0.2%, and 0.2% due to the lightweight conditional channel weighting block we designed. From the segmentation results of Network 3 and Network 6, we can obtain that our LMRF module reduced the number of network parameters by about 1M, reduced the computational effort by about 3G, and made improvements in all three brain tumor regions to some extent. The Dice coefficients of our HMNet for the three brain tumor regions were 0.781, 0.901, and 0.823, respectively. The visualization of our segmentation results obtained using the network with different methods is given in Figure 9. The results of HMNet are closer to the ground truth and have better segmentation results than the baseline, which further proves the effectiveness of our improved method in our network.

### 3.8. Comparative Experiments on the BraTS 2018 and BraTS 2019 Datasets

In order to evaluate the generalized performance of the HMNet, we compared the segmentation performance of the HMNet and other advanced networks on the BraTS 2018 and BraTS 2019 datasets.

The results on the BraTS 2018 dataset are shown in Table 9. The experiment results of 3D U-net, 3D ESP-Net, and DMF-Net are obtained by retraining the networks. The HMNet was more lightweight than the 3D U-net. The Dice coefficients of the HMNet for ET, WT, and TC were 0.786, 0.901, and 0.843, respectively. Compared to the 3D ESP-Net and DMF-Net, our proposed network had fewer parameters but higher segmentation accuracy. The Dice coefficients were higher than other networks. The Dice coefficients of HMNet were 0.9%, 0.6%, and 4.6% higher than the latest network proposed by Akbar et al. [28] in ET, WT, and CT, respectively, and the Hausdorff95 distances for ET, WT, and TC were shorter by 1.20 mm, 4.40 mm, and 1.02 mm. The HMNet is more competitive than other lightweight networks and non-lightweight networks.

Table 10 displays the experiment results on the BraTS 2019 dataset. The results of 3D U-Net, 3D ESP-Net, DMF-Net, and HDC-Net are obtained by retraining the networks. As we can see, the Dice coefficients of the HMNet for ET, WT and TC were 0.772, 0.899 and 0.830, respectively. Compared to 3D ESP-Net, the HMNet was lighter, and the Dice was higher. Compared to DMF-Net and HDC-Net, the Dice coefficients of the HMNet for ET and WT were about the same but higher for TC. Compared to the latest network proposed by Wang et al. [48], the Dice for ET was lower by 1.7%, but the Dice for TC was higher by 1.1%. In general, the network we proposed is more competitive than other networks.

## 4. Discussion

In this paper, we proposed a lightweight brain tumor segmentation network HMNet. Specifically, the shape, size and location of brain tumors vary greatly among patients, and noise is introduced during MRI scanning, resulting in blurred boundaries between tumor regions, making it difficult to segment brain tumors accurately. In addition, the networks based on 3D convolutions are computationally intensive, so these networks are not efficient enough for clinical practice. To balance segmentation accuracy and network complexity, we designed a hierarchical multi-scale brain tumor segmentation network.

We ran tests on the BraTS 2020 dataset to verify the effectiveness of the HMNet. The dice similarity coefficients of HMNet for ET, WT, and TC are 0.781, 0.901, and 0.823, respectively. Extensive experiments on the BraTS 2018, BraTS 2019, and BraTS 2020 datasets show that the proposed network has achieved satisfactory performance compared with the SOTA approaches.

Compared with the baseline network in Table 8, the parameters and flops of HMNet are significantly reduced while improving segmentation accuracy. The reason for this is that we introduce the depthwise separable convolution to replace the traditional 3 × 3 × 3 convolutions in the LCC block and LMRF module and replace the costly 1 × 1 × 1 convolution in the 3D shuffle block with CWU and SWU to overcome the problem of the heavily-used 1 × 1 × 1 convolution. HMNet has higher segmentation accuracy than the baseline network because the CWU and SWU make the network more focused on the areas associated with the tumor. In addition, the LMRF module and LCC block adopt the idea of feature reuse, where shallow feature information is directly used by deeper layers to prevent network degradation or overfitting. This can reduce the computational effort while reducing feature redundancy and improving the segmentation accuracy of the network.

Compared with other non-lightweight brain tumor segmentation networks, our proposed HMNet has better segmentation performance and lower model complexity. This is mainly because the PMRNet maintains the high-resolution features during feature propagation and ensures that the features contain accurate detailed information. However, the networks based on U-net lose some image detail information from multiple downsampling, especially at the edges of brain tumors. Moreover, our proposed HMNet is competitive with other lightweight brain tumor segmentation networks. Some lightweight networks such as DMF-Net and HDC-Net are based on U-net, so they may lose some important information. Furthermore, HDC-Net is too light to segment brain tumor boundaries accurately. Overall, our network achieves a better balance of accuracy and complexity.

## 5. Conclusions

In this paper, we propose a hierarchical multi-scale brain tumor segmentation network (HMNet), which consists of a high-resolution master branch with the highest resolution and three branches from high to low resolution, in which we can maintain the high resolution of the input image throughout the path based on parallel high-resolution feature extraction, thus preserving spatial details and performing more accurate tumor region segmentation for brain tumors, especially for fuzzy boundaries. Our network has the following advantages: (1) the multi-scale branches can provide different sensory fields for brain tumors of different sizes, which can effectively segment brain tumors of different shapes, sizes, and locations; (2) the network exchanges and fuses multi-scale features during feature propagation, which makes the extracted rich brain tumor feature; and (3) the introduction of a lightweight conditional channel weighting (LCC) block instead of the original 3D convolution and the use of the lightweight multi-resolution fusion (LMRF) module for feature fusion can greatly reduce the number of parameters and computation and improve computational efficiency. The effectiveness of the network is verified by a series of ablation experiments conducted on the BraTS 2020 dataset. Finally, the comparison with current state-of-the-art networks on the BraTS 2018 and BraTS 2019 datasets demonstrates the competitiveness of the HMNet.

## Figures and Tables

**Figure 1 jcm-12-00538-f001:**
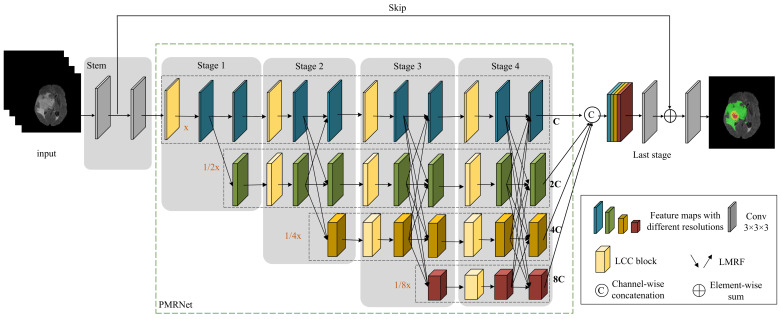
The architecture of our proposed HMNet. x denotes the input resolution, and C denotes the channel number of feature maps. We design a backbone network (PMRNet), which contains a high-resolution branch and parallel multi-resolution branches. The high-resolution branch can maintain the spatial details of the brain tumor. Meanwhile, multi-resolution feature exchange and fusion enables the receptive fields of the network to adapt to brain tumors with different sizes and shapes. In PMRNet, we propose a lightweight conditional channel weighting (LCC) block and a lightweight multi-resolution fusion (LMRF) module to overcome the large computational overhead caused by expensive 3D convolution and the fusion layers.

**Figure 2 jcm-12-00538-f002:**
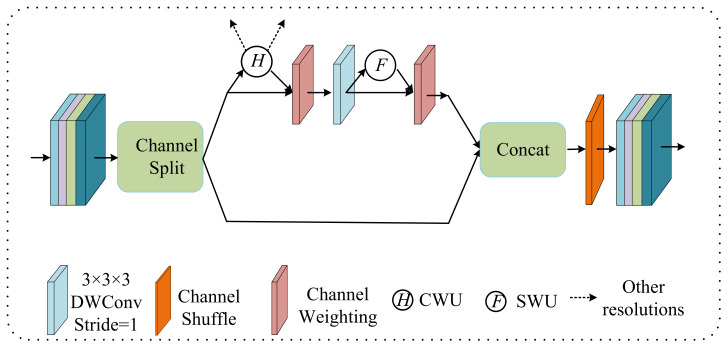
The structure of LCC block. First, the input channels are split into two branches by a channel-splitting operation. The main branch runs a channel weighting unit (CWU), 3 × 3 × 3 depthwise separable convolution (DWConv), and spatial weighting unit (SWU). The other branch is mapped as constant.

**Figure 3 jcm-12-00538-f003:**
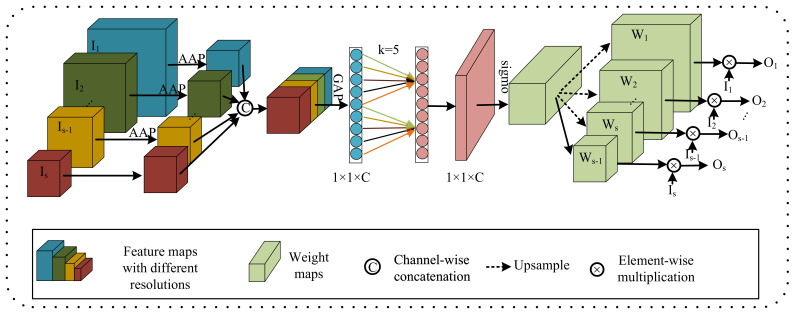
The structure of CWU. The input feature map of the parallel multi-resolution branch is subjected to the adaptive average pooling (AAP) operation to make its resolution equal to the minimum resolution of the branch and concatenated in the channel direction. Then, global average pooling (GAP) is performed to compress the spatial features of the input feature map, and the compressed feature maps are subjected to a one-dimensional convolution operation to learn the importance of different channels.

**Figure 4 jcm-12-00538-f004:**
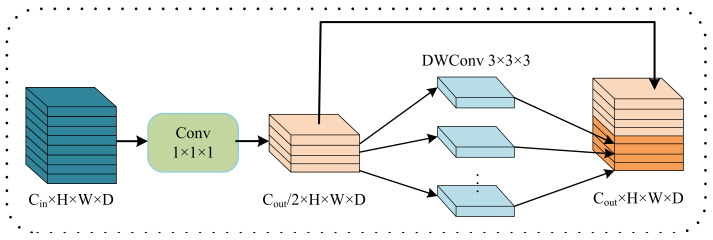
The dimensionality reduction method in LMRF module. This method can achieve the same function as 1 × 1 × 1 convolution and reduce feature redundancy.

**Figure 5 jcm-12-00538-f005:**
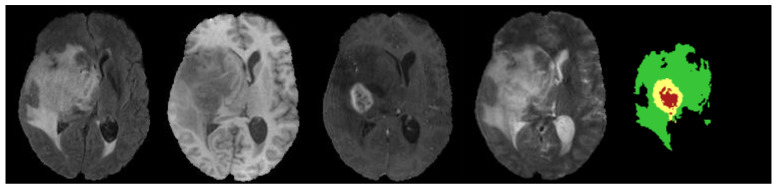
Visualization of a sample from the BraTS 2020 dataset. From left to right, there are Flair, T1, T1c and T2, respectively. The fifth image is the ground truth label. Label 1 is the necrotic and non-enhancing area in red. Label 2 is the edema area in green. Label 4 is the enhancing tumor area in yellow.

**Figure 6 jcm-12-00538-f006:**
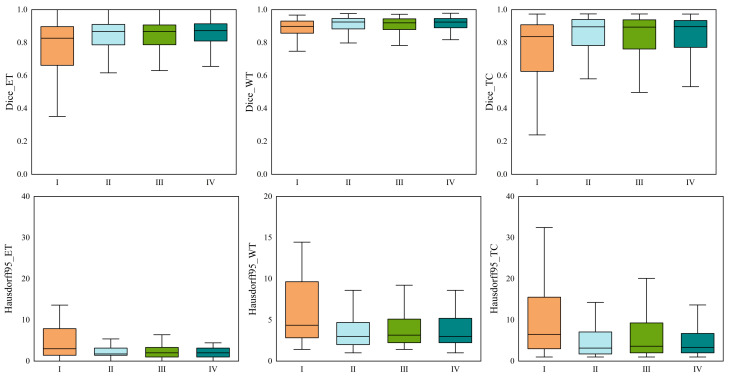
Boxplots of the Dice and Hausdorff95 distances of our proposed HMNet and other networks on the BraTS 2020 validation set. I, II, III, and IV stand for 3D U-net, DMF-Net, HDC-Net, and HMNet, respectively.

**Figure 7 jcm-12-00538-f007:**
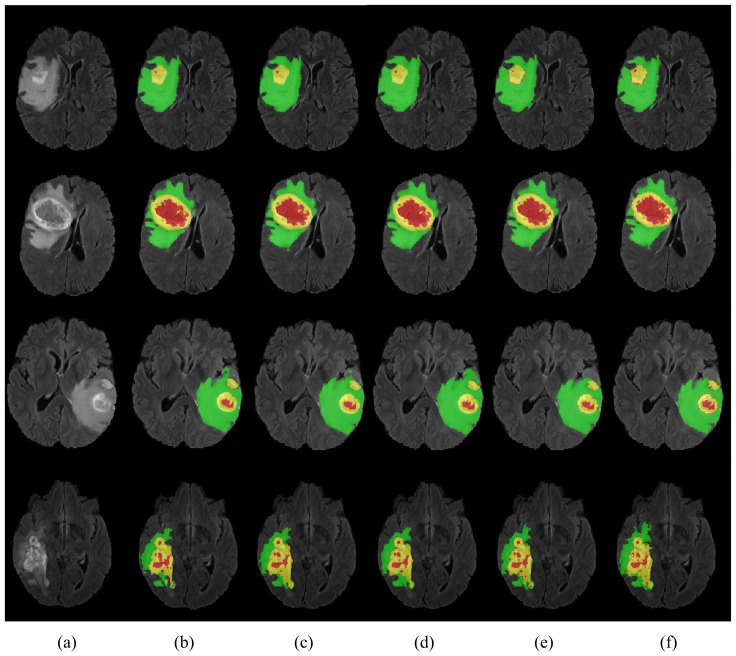
Visualization of segmentation results. Each row represents a different sample. (**a**) is the modality of Flair. (**b**–**f**) are the segmentation results of 3D Unet, DMF-Net, HDC-Net, HMNet, and ground truth. The red area is necrosis and non-enhancing, the yellow area is enhancing tumor, and the green area is edema.

**Figure 8 jcm-12-00538-f008:**
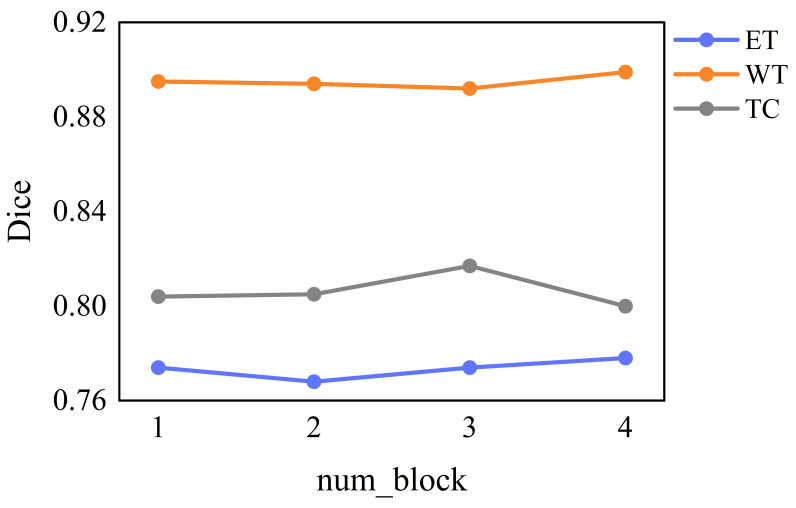
The Dice coefficients of the baseline with different num_block on the BraTS 2020 validation set.

**Figure 9 jcm-12-00538-f009:**
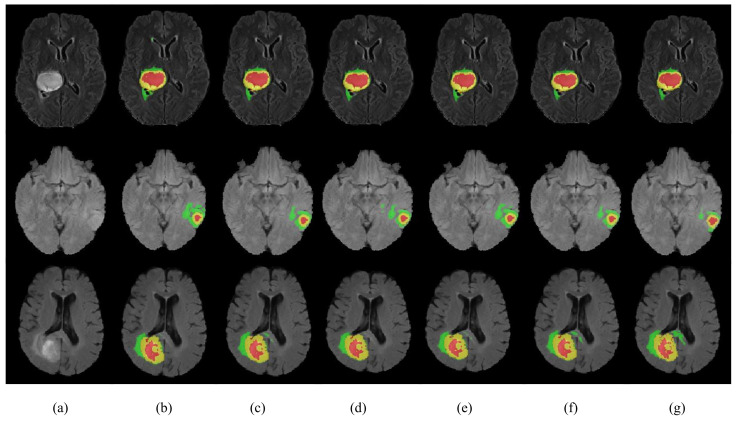
Visualization of segmentation results. Each row represents a different sample. (**a**) is the modality of Flair. (**b**–**g**) are the segmentation results of Network 1 to Network 5 in Table 8 and the ground truth. The yellow area is the enhancing tumor area. The red area is the necrosis and non-enhancing area. The green area is the edema area.

**Table 1 jcm-12-00538-t001:** The network structure of HMNet, where Conv 3 × 3 × 3 Block stands for 3 × 3 × 3 convolution, and Group Normalization, ReLU, and Concat stand for channel-wise concatenation.

Layer	Output Size	Operation	Resolution Branch	Output_Channels
Input	128 × 128 × 128			4
Stem	128 × 128 × 128	Conv 3 × 3 × 3 Block	1	32
	64 × 64 × 64	Conv 3 × 3 × 3 Block	1	32
Stage1	64 × 64 × 64	LCC block	1, 1/2	32, 64
		LMRF		
Stage2	64 × 64 × 64	LCC block	1, 1/2, 1/4	32, 64, 128
		LMRF		
Stage3	64 × 64 × 64	LCC block	1, 1/2, 1/4, 1/8	32, 64, 128, 256
		LMRF		
Stage4	64 × 64 × 64	LCC block	1, 1/2, 1/4, 1/8	32, 64, 128, 256
		LMRF		
Skip	128 × 128 × 128	Element-wise sum		32
Last Stage	64 × 64 × 64	Upsample, Concat		480
	64 × 64 × 64	Conv 3 × 3 × 3 Block		32
	128 × 128 × 128	Upsample		32
	128 × 128 × 128	Conv 3 × 3 × 3 Block		3

**Table 2 jcm-12-00538-t002:** The structure of LCC block. Take the first LCC block in Stage 2 as an example.

Input Size	Input Channels	Details	Output Size	Output Channels
64 × 64 × 64	32	Channel Split	64 × 64 × 64	16, 16
64 × 64 × 64	16	Adaptive average pooling	32 × 32 × 32	16
32 × 32 × 32	64	Concat	32 × 32 × 32	80
32 × 32 × 32	16			
32 × 32 × 32	80	Conv 1 × 1	1 × 1 × 1	80
1 × 1 × 1	80	Dimension extension, Upsample	64 × 64 × 64	16
64 × 64 × 64	16	DWConv 3 × 3 × 3	64 × 64 × 64	16
64 × 64 × 64	16	Global Average Pooling	1 × 1 × 1	16
1 × 1 × 1	16	Conv 1 × 1 × 1, Conv 1 × 1 × 1	64 × 64 × 64	16
64 × 64 × 64	16	Sigmoid, Dot product	64 × 64 × 64	16
64 × 64 × 64	16	Concat	64 × 64 × 64	32
64 × 64 × 64	16			
64 × 64 × 64	32	Channel Shuffle	64 × 64 × 64	32

**Table 3 jcm-12-00538-t003:** The structure of LRMF module.

Method	Input Size	Input Channels	Details	Output Size	Output Channels
Downsample	32 × 32 × 32	64	DWConv 3 × 3 × 3	16 × 16 × 16	128
Upsample	32 × 32 × 32	64	Upsample	64 × 64 × 64	64
	64 × 64 × 64	64	Conv 1 × 1 × 1	64 × 64 × 64	16
	64 × 64 × 64	16	DWConv 3 × 3 × 3	64 × 64 × 64	16
	64 × 64 × 64	16	Concat	64 × 64 × 64	32
	64 × 64 × 64	16			

**Table 4 jcm-12-00538-t004:** Comparison with other non-lightweight brain tumor segmentation networks on the BraTS 2020 validation set. A (-) denotes that the results are not reported.

Network	Params/M	Flops/G	Dice			Hausdorff95/mm		
			ET	WT	TC	ET	WT	TC
3D U-net [24]	16.21	1669.5	0.706	0.862	0.737	39.774	12.720	19.106
Robin et al. [46]	12.42	296.82	0.768	0.891	0.819	38.351	6.320	7.345
Ghaffari et al. [47]	-	-	0.780	0.900	0.820	**7.710**	5.140	**6.640**
Wang et al. [48]	-	-	0.787	0.900	0.817	17.947	**4.964**	9.769
Raza et al. [49]	-	-	**0.800**	0.860	0.822	-	-	-
HMNet (ours)	0.80	129.4	0.781	**0.901**	**0.823**	21.340	5.954	7.055

**Table 5 jcm-12-00538-t005:** Comparison with other lightweight brain tumor segmentation networks on the BraTS 2020 validation set.

Network	Params/M	Flops/G	Dice			Hausdorff95/mm		
			ET	WT	TC	ET	WT	TC
3D ESP-Net [50]	3.36	76.51	0.690	0.871	0.786	31.299	7.100	14.617
DMF-Net [40]	3.88	27.04	**0.783**	0.899	0.817	26.255	**5.099**	11.515
HDC-Net [51]	**0.29**	**25.6**	0.767	0.896	0.797	32.435	5.476	14.203
HMNet (ours)	0.80	129.4	0.781	**0.901**	**0.823**	**21.340**	5.954	**7.055**

**Table 6 jcm-12-00538-t006:** The evaluation metrics of baseline using different num_block.

Network	Num_Block	Params/M	Flops/G	Dice		
				ET	WT	TC
1	×1	6.87	164.4	0.774	0.895	0.804
2	×2	7.29	168.5	0.768	0.894	0.805
3	×3	7.74	173.8	0.774	0.892	0.817
4	×4	8.17	178.5	0.778	0.899	0.800

**Table 7 jcm-12-00538-t007:** The evaluation metrics of baseline using different methods. Here, baseline_W32 and baseline_W64 represent networks with PMRNet input size of 32 × 32 × 32 and 64 × 64 × 64, respectively, and without Skip represents the removal of the Skip operation. Avg_Dice refers to the average Dice of the three tumor regions.

Method	Avg_Dice	Dice		
		ET	WT	TC
baseline_W32(without Skip)	0.810	0.720	0.896	0.815
baseline_W32	0.812	0.738	0.878	0.820
baseline_W64(without Skip)	0.821	0.748	0.896	0.821
baseline_W64	0.824	0.774	0.895	0.804

**Table 8 jcm-12-00538-t008:** The evaluation metrics for networks using different modules.

Network	Method	Param/M	Flops/G	Dice			Hausdorff95/mm		
				ET	WT	TC	ET	WT	TC
1	baseline	6.87	164.4	0.774	0.895	0.804	21.680	7.854	14.138
2	baseline+Shuffle	0.96	134.2	0.763	0.898	0.810	22.927	5.989	13.457
3	baseline+Shuffle+LMRF	0.85	130.3	0.780	0.897	0.820	27.071	6.536	7.539
4	baseline+LCC	0.92	133.3	0.772	0.900	0.812	27.061	6.024	10.329
5	HMNet(ours)	**0.80**	**129.4**	**0.781**	**0.901**	**0.823**	**21.340**	**5.954**	**7.055**

**Table 9 jcm-12-00538-t009:** Comparison with other brain tumor segmentation networks on the BraTS 2018 validation set. Here, (-) denotes that the results are not reported.

Network	Params/M	Flops/G	Dice			Hausdorff95/mm		
			ET	WT	TC	ET	WT	TC
3D U-net	16.21	1669.5	0.759	0.885	0.717	6.040	17.100	11.620
3D ESP-Net [50]	3.36	76.51	0.737	0.883	0.814	5.302	5.463	7.853
DMF-Net [40]	3.88	27.04	0.781	0.899	0.835	3.385	4.861	7.743
Akbar et al. [28]	-	-	0.777	0.895	0.797	3.90	9.13	8.67
HMNet (ours)	0.80	129.4	0.786	0.901	0.843	2.699	4.727	7.731

**Table 10 jcm-12-00538-t010:** Comparison with other brain tumor segmentation networks on the BraTS 2019 validation set. Here, (-) denotes that the results are not reported.

Network	Params/M	Flops/G	Dice			Hausdorff95/mm		
			ET	WT	TC	ET	WT	TC
3D U-net	16.21	1669.5	0.737	0.894	0.807	6.41	12.32	10.44
3D ESP-Net [50]	3.36	76.51	0.663	0.871	0.786	6.84	7.42	9.74
DMF-Net [40]	3.88	27.04	0.776	0.900	0.815	2.99	4.64	6.22
HDC-Net [51]	0.29	25.6	0.773	0.893	0.818	4.19	6.74	7.98
Wang et al. [48]	-	-	0.789	0.900	0.819	3.73	5.64	6.04
HMNet (ours)	0.80	129.4	0.772	0.899	0.830	4.01	5.21	6.57

## Data Availability

Data will be made available on request.

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
