# Peer review of "HMNet: Hierarchical Multi-Scale Brain Tumor Segmentation Network"

_jcm, 2023, doi:10.3390/jcm12020538_

Round 1
Reviewer 1 Report
It is good work. However, authors need to validate the performance metrics in a numerical way as well.
Proof read the paper for minor typos and English language usage
Reviewer 2 Report
In my opinion, this is a very good and interesting article. However, it would be good to make some improvements. I have the following comments:
1. The structure of the article deviates from that recommended by the template. It is worth to improve it.
2. There is a lack of discussion. The article answers the "what" question, but the consideration of "why" is missing.
3. The authors used a very complex neural network model. However, they did not write how it was created. Was it the result of testing several models (or other sets of model parameters)? If so, the results of these tests and alternative models should be presented. Was the model adopted just like that, without deeper analysis? If so, model optimization (e.g., selection of other model structures or metaparameters) should be performed.
4. Regardless of the figures and description, a listing with a Pytorch model of neural network can be used.
5. It would be very valuable to include a link to the source code (GitHub / GitLab) so that the experiment can be repeated.
